# SCORE-BASED NEURAL PROCESSES

## ABSTRACT

Neural processes (NP) are a flexible class of models that generate stochastic processes by operating on finite-dimensional marginal distributions. NPs are designed to maintain exchangeability and marginal consistency, which are necessary to define a valid stochastic process. However, NP variants can come with drawbacks such as limited expressivity, uncorrelated samples, and consistency sacrifices. To address the issues of previous NPs, we introduce score-based neural processes, *scoreNP*, which incorporate a score-based generative model within the neural process paradigm. This score-based approach enhances expressivity, allowing the model to capture complex non-Gaussian distributions of functions, generate correlated samples, and maintain marginal consistency. Previously, no NP variant has been able to maintain conditional consistency. We show that using *guidance* methods from conditional diffusion sampling, *scoreNP* is the first NP is able to satisfy conditional consistency. Empirically, *scoreNP* performs well qualitatively and quantitatively well across a range of unconditional and conditional functional generation tasks.

## 1 INTRODUCTION

Neural processes (NP) (Garnelo et al., 2018) are a class of generative models on function spaces, that is they model stochastic processes. Rather than operating directly in the function space NPs leverage the Kolomogorov Extension theorem (KET) to operate on the finite dimensional marginal distributions. They can be used for a range of tasks including function regression and Bayesian optimisation. In practical settings, such as weather monitoring (Andersson et al., 2023), noisy and irregularly gathered data are often the norm. Moreover, the functional nature of these settings motivates the use of NPs to provide a functional probabilistic perspective. However, the original NP (Garnelo et al., 2018) lacks expressivity, limiting its applicability. Follow-up works, including Attentive NPs (ANPs) (Kim et al., 2019), Convolutional NPs (ConvNPs) (Gordon et al., 2020) and Gaussian NPs (GNPs) (Bruinsma et al., 2021) aim to improve expressivity. However, these approaches fall into the setting of predictive NPs, meaning the generative process no longer follows an underlying stochastic process.

Recently, Markov NPs (MNPs) (Xu et al., 2023) were proposed as a generalisation of the original NP, offering greater expressivity while maintaining the generative nature through a finite hierarchical model of NPs. Score-based generative models are an expressive class generative models and can be interpreted as an infinitely deep hierarchical model which motivates further generalisation to incorporate them within the NP framework. Recently, (Geometric) Neural Diffusion Processes (NDPs) (Mathieu et al., 2023; Dutordoir et al., 2023) adopted the NP setting for diffusion models. However, due to the architectural constraints, they disregard the marginal consistency element of the KET, (Eq.2), meaning they do not model valid stochastic processes.

Score-based generative models (Song et al., 2021b) have become one of the most commonly used class of diffusion models. Typically, diffusion models are applied in the finite dimensional domain such as audio or images. Recently, they have been extended to work with the infinite-dimensional function space (Pidstrigach

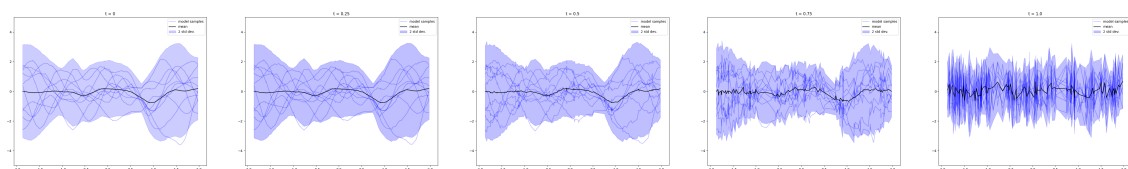

Figure 1: ScoreNP unconditional generation: Denoising from $\mathcal{N}(0, I)$ (right to left) to a distribution over functions drawn from a Matern-$(\frac{5}{2})$ kernel

et al., 2023; Bond-Taylor & Willcocks, 2023; Phillips et al., 2022; Biloš et al., 2023; Franzese et al., 2023). While these works extend diffusion models to function spaces, operating directly on the function space means practical implementations sacrifice consistency due to discretisation. Moreover, they require full regularly spaced functional data, which is not always feasible.

Our main contribution is the introduction of a novel NP framework, *score-based neural processes* (scoreNP). This extends generative neural processes by incorporating score-based generative models through a joint diffusion process in a latent-data space, which satisfies the KET consistency conditions. The score-based approach enables us to model expressive distributions and maintain strong performance even in low or zero-context settings. Additionally, it allows us to train the model using a variational adaptation of the score-matching objective, which we demonstrate can be interpreted as a variational lower bound on the log-likelihood.

Additionally, unlike all existing NP models we show how utilising a form of *guidance* on an unconditional diffusion model to produces posterior distributions which are consistent *w.r.t* to both the context and target set. This defines a unique stochastic process over the in domain. It also enables us to generate both unconditional (see Fig.1) and conditional samples from a single objective and training run.

We validate our method experimentally through 1D functional regression tasks which include complex non-Gaussian distributions.

## 2 BACKGROUND

In this section we will discuss requirements for modelling stochastic processes and the necessary backgrounds of Neural processes and score-based generative models.

### 2.1 STOCHASTIC PROCESSES

A Stochastic Process (SP) is defined as a set of random variables $\{y_i\}_{i \in I}$, where $I$ is an (uncountable) index set, where $y_i$ take values in $Y$. This can be seen as a map $F : I \to Y$, and $p(F(x_i)) = p(y_i)$ is a probability distribution. By Kolomogorov's Extension theorem (KET), one can uniquely define a stochastic process with uncountable index set, e.g $I = \mathbb{R}$, if for all finite collections $U \subset I$, the marginal distributions are both *(marginally) consistent* and *exchangeable* (permutation invariant). More specifically a distribution $p$ is exchangeable if given any permutation $\pi \in \mathbb{S}^n$ then;

$$p(y_{1:n}|x_{1:n}) = p(y_{\pi(1:n)}|x_{\pi(1:n)}) = p(y_{\pi(1)}, \ldots, y_{\pi(n)}|x_{\pi(1)}, \ldots, x_{\pi(n)}), \quad (1)$$

and (marginally) consistent if, given any integers $m < n$, if we marginalise the distribution across the marginal densities of indices $m + 1 : n$, the following distributions are equal.

$$p(y_{1:m}|x_{1:m}) = \int_{y_{m+1:n}} p(y_{1:n}|x_{1:n}) dy_{m+1:n} \quad (2)$$

There is also another requirement for valid stochastic processes under a conditioning context set, that is conditional consistency. For any context set $\mathcal{C} = \{x_{1:m}, y_{1:m}\}$ and a prior $p(f)$ on a stochastic process $f$.

$$p(f) = \int p(f|\mathcal{C})p(y_{1:m}|x_{1:m})dy_{1:m} \tag{3}$$

## 2.2 Neural Processes

Neural Processes (Garnelo et al., 2018) are flexible models for probabilistic modelling in the function space. However, their original formulation comes with certain limitations such as underfitting and limited expressivity. Follow up works build on NPs, Convolutional NPs, (Gordon et al., 2020), Attentive NPs (Kim et al., 2019), Markov NPs (Xu et al., 2023), to alleviate these shortcomings.

Based on de Finetti's theorem, NPs (Garnelo et al., 2018) construct the following generative model:

$$p(y_{1:n}) = \int q(z) \prod_{i=1}^{n} \mathcal{N}(y_i; \mu_\theta(x_i, z), \sigma_\theta(x_i, z))dz \tag{4}$$

Where $q(z)$ is assumed to be a tractable prior such as $\mathcal{N}(0, I)$ which regularises a variational posterior $q_\phi(\cdot|(x_{1:n}, y_{1:n}))$ and $\mu_\theta, \sigma_\theta$ are neural networks which capture the complexities of the model. They representa $z$ with a high-dimensional random vector which acts as a global representation of functions from underlying SP $f$. Sampling from Eq.4 gives finite marginal distributions which satisfy the KET consistency conditions. However, the Gaussian assumption on $y_i$ and the single step decoding $(z \rightarrow y)$ limits the expressive power of this model, especially due to the constraints imposed on the neural network architectures by the KET conditions. MNPs (Xu et al., 2023) propose a generalisation to Eq.4.

$$p_\theta(y_{1:n}|x_{1:n}) = \int_{z^{(1:k)}} \int_{y_{1:n}^{(1:k)}} q(z^{(1:k)})p(y_{1:n}^{(0)}) \prod_{t=1}^{k} \prod_{i=1}^{n} p_\theta^t(y_i^{(t)}|y_i^{(t-1)}, x_i, z^{(t)})dz^{(1:k)} \tag{5}$$

Where $y_{1:n}^{(0)}$ are sampled from a tractable simple prior and $p_\theta^{(t)}$ is a normalising flow parameterised by a neural network. This generalisation builds a finite markov chain of NPs for expressive SPs.

We develop on this insight by operating in the continuous-time domain building an infinite markov chain of NPs via a denoising diffusion model.

NPs can be split into two classes; latent and conditional. *ScoreNP* falls into the former. Latent NPs hold certain advantages over conditional NPs such as the full generative nature and correlated individual samples. This allows them to be applied to settings with changing context such as Bayesian optimisation. When referring to NPs (Garnelo et al., 2018) we will be assuming the latent NP set up, inline with previous literature.

## 2.3 Score-Based Generative Models

*Score-based generative models* (SGM) (Song et al., 2021b) are a class of diffusion models where the noising and denoising process is determined by continuous-time stochastic differential equations (SDE). The forward and reverse SDEs are defined $\forall \tau \in [0, 1]$. The forward process is defined by an Itô SDE;

$$dy^\tau = f(y, \tau)d\tau + g(\tau)dB^\tau; \tag{6}$$

Where $f(\cdot)$ and $g(\cdot)$ are affine *drift* and *diffusion* coefficients respectively and $B$ is a standard Wiener process. Where $p^\tau(y)$ is the probability density of $y^\tau$ and Eq.6 is designed such that $y^1 \sim p^1$ is a tractable probability distribution. The reverse of the Itô SDE Eq.6 is also an Itô SDE operating backwards in time, and is given by;

$$dy_\tau = \left[ f(y, \tau) - g(\tau)^2 \nabla_{y^\tau} \log p^\tau(y) \right] d\tau + g(\tau)d\bar{B}^\tau \tag{7}$$

$\bar{B}^\tau$ is a standard Wiener process under time reversal. $\nabla_{y^\tau} \log p^\tau(y)$ is called the *score*. If the *score* is known for all $\tau$ we can solve Eq.7 to generate samples from $p^0(y)$. However, it is intractable for arbitrary probability distributions, hence it is estimated by a Score Network, $S_\theta(y^\tau, \tau)$ which is approximated through a Score Matching (SM) objective:

$$\text{SM}(\theta) := \int_0^1 \mathbb{E}_{p^\tau(y)} \left[ \omega(\tau) || \nabla_{y^\tau} \log p^\tau(y) - S_\theta(y^\tau, \tau) ||^2 \right] d\tau \tag{8}$$

where $\omega(\tau)$ is a time-dependent scalar weighting function. $p^\tau(y)$ is intractable, however the transition density $p^{\tau|0} = p(y^\tau|y^0)$ is known for affine $f$ and $g$.

**(Denoising) Score Matching** is a tractable equivalent objective for estimating the the score of a distribution $\nabla_y \log p(y)$ (Hyvarinen, 2005) based on the transition density.

$$\text{DSM}(\theta) := \int_0^1 \mathbb{E}_{p^0(y)} \mathbb{E}_{p^{\tau|0}(y)} \left[ \omega(\tau) || \nabla_{y^\tau} \log p^{\tau|0}(y) - S_\theta(y^\tau, \tau) ||^2 \right] d\tau \tag{9}$$

## 3 SCORE-BASED NEURAL PROCESS (SCORENP)

In this section we introduce our model, Score-Based Neural Processes (ScoreNP) which operates a score-based generative model jointly in the data-space and an encoded latent-space. We first define the time-dependent marginal distributions and the generative process before discussing how the model is parameterised and trained.

At the high-level we can consider our model as an infinitely deep hierarchical NP, similar to how diffusion models are interpreted as infinitely deep VAEs (Huang et al., 2021). As in (latent) NPs, we encode the function input and outputs $(x_{1:n}, y_{1:n})$ into a high-dimensional latent vector $z$, using a variational encoder $q_\phi$, which acts as a representation for the function. Function outputs $y_{1:n}$ and $z$ are jointly diffused under a forward process. A reverse diffusion step at $\tau$ for a single target point $y_i^\tau$ conditions on itself, the noised latent $z^\tau$, and the input location $x_i$, which enforces a conditional independence on $y_i$'s. A denoising step for $z^\tau$ is treated as a typical diffusion

---

**Algorithm 1** Single Training Step for ScoreNP

**Require:** $(x_{1:n}, y_{1:n})$, $q_\phi(\cdot)$, $\tau \sim U(0,1)$, $S_{\theta,\psi}(\cdot)$, $p^{\tau|0}(\cdot)$
  $z \sim q_\phi(\cdot|(x_{1:n}, y_{1:n}))$
  $z^\tau \sim p^{\tau|0}(z)$
  Compute latent score: $s_\psi(z^\tau, \tau)$
  **for** $i \in [1:n]$ **do**
    $y_i^\tau \sim p^{\tau|0}(y_i)$
    Compute conditional data score: $s_\theta(y_i^\tau, x_i, z^\tau, \tau)$
    $s_i = s_\theta(y_i^\tau, x_i, z^\tau, \tau)$
  **end for**
  $S_\theta(y^\tau, x, z^\tau, \tau) = \{s_1, \ldots, s_n\}$
  Compute $\mathcal{L}(\cdot)$          ▷ Eq.20
  Get gradients
  Update $\{\theta, \psi, \phi\}$

---

denoising step. Learning a prior over the latent space with a diffusion produces samples from well-supported regions which improves performance in low context domains. The conditional independence provides a generative model for valid SPs. The encoder, $q_\phi$ is optimised towards improving the target denoising steps.

### 3.1 GENERATIVE PROCESS OF SCORENP

We propose a time-dependent joint distribution which is defined for all $\tau \in [0, 1]$ and under the generative model ensure KET conditions are satisfied. Based on the conditional de Finettis theorem we define a time-dependent joint distributions as:

$$p^\tau(y_{1:n}, z | x_{1:n}) = p^\tau(z) \prod_{i=1}^n p^\tau(y_i | z, x_i) \tag{10}$$

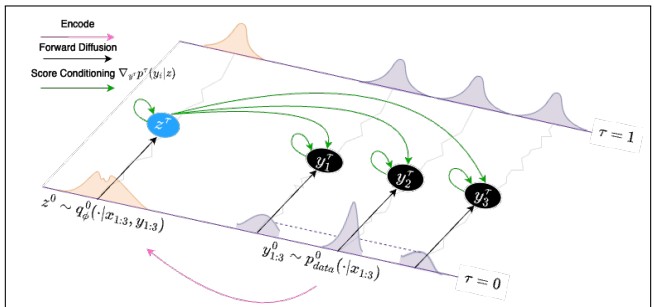
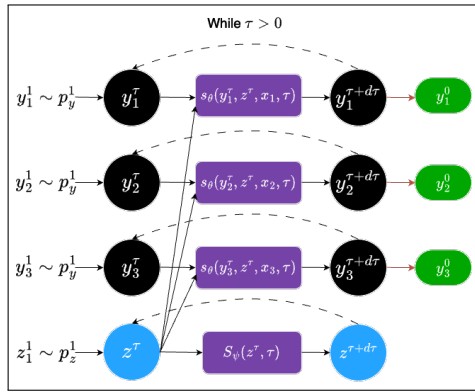

(a) Training Step: visually show the joint diffusion process under a training step, **(1)** a data point is sampled and encoded into a latent variable $z^0 \sim q(\cdot|(x,y))$. **(2)** Sample $\tau \sim \mathcal{U}(0,1)$ and apply a forward diffusion on $(z^0, y_{1:3}^0)$ to get $(z^\tau, y_{1:3}^\tau)$. **(3)** Approximate the score $\nabla_{y_{1:n}^\tau, z^\tau} \log p^\tau(z) \prod_{i=1}^n p^\tau(y_i|z,x_i)$ at $\tau$ using the score network $S_{\theta,\phi}(y^\tau, z^\tau, \tau)$ and reparameterise to update the encoder. see Alg.1

(b) A graphical model of the Generative process under sampling. We get initial states $(y_{1:3}^1, z^1)$ by sampling from the tractable distributions $p_z^1, p_y^1$. Compute the *scores* with dependence indicated with arrows. The state is updated through the scores and the SDE change in time $d\tau$. This process is iteratively (dotted lines) applied while $\tau > 0$. Finally, we set the current states $y_{1:n}^0$ and return targets (see Alg.3)
.

Figure 2: Visualisation of training step *(left)* and graphical model of generative process *(right)*

Where $p^0(y_{1:n}, z|x_{1:n})$ represents the distribution of the marginals from the underlying SP. The similarities to the NP generative model (Eq.4) implies marginalising over the latent $z$ on $p^0(y_{1:n}, z|x_{1:n})$ gives valid finite marginal distribution. Hence, we want to sample from $p^0(z) \prod_{i=1}^n p^0(y_{1:n}, z|x_{1:n})$. Extending the forward diffusion process Eq.6, to the joint variable $u = (y_{1:n}, z \mid x_{1:n})$ we get;

$$du^\tau = f(u,\tau)d\tau + g(\tau)dB^\tau \tag{11}$$

Using Bayes' rule we get two equivalent forms for the reverse SDE;

$$du^\tau = \left[ f(u,\tau) - g(\tau)^2 \nabla_{y_{1:n}^\tau, z^\tau} \log p^\tau(y_{1:n} \mid z) q^\tau(z) \right] d\tau + g(t)d\bar{B}^\tau \tag{12}$$

$$du^\tau = \left[ f(u,\tau) - g(\tau)^2 \nabla_{y_{1:n}^\tau, z^\tau} \log p^\tau(y_{1:n}) q^\tau(z \mid y_{1:n}) \right] d\tau + g(t)d\bar{B}^\tau \tag{13}$$

Noting that Eq.12 has the score of the time-dependent marginals before invoking conditional independence, we can rewrite is to be the reverse diffusion process which depends on the *score* of the time-dependent marginals Eq.10

$$du^\tau = \left[ f(u,\tau) - g(\tau)^2 \nabla_{y_{1:n}^\tau, z^\tau} \log p^\tau(z) \prod_{i=1}^n p^\tau(y_i|z,x_i) \right] d\tau + g(t)d\bar{B}^\tau \tag{14}$$

Solving the SDE Eq.14 allows us to sample from the time-dependent marginals at any point. The forward process ensures that $p^1(z)$ and $p^1(y_{1:n}|z, x_{1:n})$ are tractable distributions, namely $\mathcal{N}(0, I)$, hence solving the reverse SDE provides us with a consistent generative process for $y_{1:n}^\tau$. At $\tau = 0$ we have a generative process for the target stochastic process.

The *score* in Eq.14 is parameterised with a score network $S_{\theta,\psi}$ in the following manner to ensure marginal consistency. We have two time-dependent score models; the *data score*, $S_\theta(y_{1:n}^\tau, x_{1:n}, z^\tau, \tau) \approx \nabla_{y_{1:n}^\tau} \log \prod_{i=1}^n p^\tau(y_i|z, x_i)$ and the *latent score*, $S_\psi(z^\tau, \tau) \approx \nabla_{z^\tau} \log q^\tau(z|y_{1:n})$.

Due to the assumption that $y_i$ is conditionally independent of $y_j$ given $z$, the data score is separated into pointwise components:

$$S_\theta(y_{1:n}^\tau, x_{1:n}, z^\tau, \tau) = \begin{pmatrix} s_\theta(y_1^\tau, x_1, z^\tau, \tau) \\ s_\theta(y_2^\tau, x_2, z^\tau, \tau) \\ \vdots \\ s_\theta(y_n^\tau, x_3, z^\tau, \tau) \end{pmatrix} \approx \begin{pmatrix} \nabla_{y_1^\tau} \log p^\tau(y_1|z, x_1) \\ \nabla_{y_2^\tau} \log p^\tau(y_2|z, x_2) \\ \vdots \\ \nabla_{y_n^\tau} \log p^\tau(y_n|z, x_n) \end{pmatrix} \quad (15)$$

This ensures marginal consistency is maintained during neural network forward passes. The *latent score* $S_\psi(z, \tau)$ has no constraints placed on it and follows existing architectures for score networks.

**Theorem 1.** *Given initial value conditions that $z^1 \sim p^1(z)$ is tractable and $y_{1:n}^1 \sim p^1(y_{1:n}|x_{1:n})$ is tractable and defines a valid stochastic process over $X$, then $\forall \tau \in [0, 1]$ the probability distribution $p^\tau(z)p^\tau(y_{1:n}|z, x_{1:n})$ determined by Eq.14 is a valid stochastic process over $X$. (App.A for sketch proof)*

Thm.1 allows us to define our generative model for consistent marginals solving the reverse SDE Eq.14 and marginalising over the joint diffusion paths.

## 3.2 JOINT VARIATIONAL SCORE-MATCHING LOSS

Now that we have defined the generative model, we need to show how to train this via a score matching objective. Neural Processes Garnelo et al. (2018) maximise the log-likelihood of the marginal distribution of a target sets $(y_{1:n}, x_{1:n}) \in D_{train}$ via the Evidence Lower Bound (ELBO).

$$\log p_\theta(y_{1:n}|x_{1:n}) \geq \mathbb{E}_{q_\phi(z|y_{1:n}, x_{1:n})} [\log p_\theta(y_{1:n}|z, x_{1:n}) + \log q_\phi(z) - \log q_\phi(z|x_{1:n}, y_{1:n})] \quad (16)$$

where $p_\theta$ is the decoder based on $\mu_\theta, \sigma_\theta$ from Eq.4. Additionally, it is often desired to maximise the log-likelihood of targets *w.r.t* context set $(y_{1:m}, x_{1:m})$ where $m < n$. We avoid this setting in our work as we condition on contexts by sampling from a posterior distribution utilisng reconstruction guidance on the trained unconditional model. Replacing, the first two terms in Eq.17 with the parameterised time-dependent marginals at $\tau = 0$

$$\log p_\theta(y_{1:n}|x_{1:n}) \geq \mathbb{E}_{q_\phi(z|y_{1:n}, x_{1:n})} \left[ \log p_\theta^0(y_{1:n}|z, x_{1:n}) + \log p_\psi^0(z) - \log q_\phi(z|x_{1:n}, y_{1:n}) \right] \quad (17)$$

We get an ELBO based on the joint marginal distributions from scoreNP. However, it is infeasible to get $\log p_\theta^0(y_{1:n}|z, x_{1:n}) + \log p_\phi^0(z)$ directly. Hence, similar to (Song et al., 2021a; Huang et al., 2021) we show that with the *likelihood weighting*, $\omega(\tau) = \frac{g(\tau)^2}{2}$ we can derive an upper bound on the negative log-likelihood (NLL) through a variational score matching objective on the joint reverse process. Firstly, we define variational score-matching with *likelihood weighting*, Using Eq.18 we are able to derive an upper bound on NLL.

$$J^{\text{var-sm}}(\theta, \psi, \phi; g(\cdot)^2) := \int_0^1 \mathbb{E}_{p_{\text{data}}(y_{1:n})} \mathbb{E}_{q_\phi(z|y_{1:n})} \left[ \frac{g(\tau)^2}{2} ||\nabla_{y^\tau, z^\tau} \log p^\tau(y_{1:n}) q_\phi^\tau(z|y_{1:n}) - S_{\theta, \psi}(y^\tau, z^\tau, \tau)||^2 \right] d\tau \quad (18)$$

**Theorem 2** (Variational Lower Bound). *Let $p_{\theta, \psi}^0(y_{1:n}, z)$ be defined as the the probability joint density under reverse process at $\tau = 0$ under the reverse SDE Eq.10 and $p_\theta^0(y_{1:n})$ be the corresponding marginal: Let $p(y_{1:n}, z) = p_{data}(y_{1:n}) q_\phi(z|y_{1:n})$ be the joint data-latent distribution. Suppose, $\{(y_{1:n}^\tau, z^\tau)\}_{\tau \in [0,1]}$ is a stochastic process defined by the SDE Eq.11 with $\{(y_{1:n}^0, z^0)\} \sim p_{data}(y_{1:n}) q_\phi(z|y_{1:n})$. Then, under mild regularity conditions. (see app.A for proof)*

$$-\mathbb{E}_{p_{data}(y_{1:n})} \left[ \log p_\theta^0(y_{1:n}) \right] \leq J^{var-sm}(\theta, \psi, \phi; g(\cdot)^2) + \mathcal{H}_\phi(z^0) \quad (19)$$

This result is an extension from previous VLB's on diffusion models (Huang et al., 2021; Vahdat et al., 2021). It notably takes into account the use of Bayes' rule to allow for tractable training on the joint and the use of a diffusion model in the data space as a decoder.

Again we note that the variational Score Matching objective Eq.18 equal to a variational Denoising Score Matching objective up to a constant. Replacing, $J^{\text{var}}(\theta, \psi, \phi; g(\cdot)^2)$ in Eq.28 with its DSM equivalent we get a tractable objective.

$$\mathcal{L}(y_{1:n}, x_{1:n}; \theta, \psi, \phi) = J^{\text{var-dsm}}(\theta, \psi, \phi; g(\cdot)^2) + \mathcal{H}_\phi(z) \tag{20}$$

The parameters of the score models, $\{\theta, \psi\}$ are optimised for the objective Eq.20. Generally, we backpropagate through both score models[1] to the encoder via the reparameterisation trick (Kingma & Welling, 2022) to update the variational parameters $\phi$.

**Encoder**    The encoder, $q_\phi(z|x_{1:n}, y_{1:n})$ is parameterised using a *permutation invariant* network such as a DeepSet (Zaheer et al., 2018) or SetTransformer (Lee et al., 2019) to maintain *exchangeability* (Eq.1) as is standard in NP literature. However, since our generative process drops the encoder during sampling (Sec. 4), this is not strictly necessary. We retain *permutation invariant* encoders in this work, but the increased flexibility opens the door for future research into more expressive encoders within the *scoreNP* framework.

**Encoder Collapse** We adopt a similar training regime to LSGM (Vahdat et al., 2021) where we set $\nabla_{z^\tau} \log q^\tau(z) = -z^\tau$ initially for a pretraining stage, which regularises $q_\phi(z)$ and allows the encoder to learn a structured latent space before learning the *latent score*. Without this we found the model to suffers from encoder collapse and set $q_\phi(z) \approx \delta(0)$. In the second stage we set $\nabla_{z^\tau} \log q^\tau(z|y_{1:n}, x_{1:n}) = -\alpha z^\tau + (1 - \alpha)S_\psi(z^\tau, \tau)$, where $\alpha \in [0, 1]$ can be a learnable or explicitly defined. The encoder is either frozen or continues evolving through end-to-end learning during the second stage. Whilst, the latter is preferred we found that for certain datasets freezing the encoder was helpful to prevent encoder collapse.

**Geometric variance preserving SDE**    We apply the Geometric Variance Preserving SDE (Geo. VPSDE) from (Vahdat et al., 2021), which offers a more gradual noising process compared to the original VPSDE (Song et al., 2021b). While the reduced variance during training with *likelihood weighting* motivated this choice, we also found that the slower noising process aids in training the encoder by allowing more time with distinguishable data, making reparameterization more effective. Details of the exact SDE formulation can be found in Sec.E.3

## 4    CONDITIONAL GENERATION

In many generative modelling scenarios we want to be to generate samples from a posterior distribution conditioning on some input, e.g inpainting, class-conditional generation. This is the case for the Neural Process Family, where we want to generate a posterior distribution over a function space given a (finite) *context set* $\mathcal{C} = \{x_\mathcal{C}, y_\mathcal{C}\}$, and this is achieved by modelling a posterior distribution over *target sets*, $\mathcal{T} = \{x_\mathcal{T}, y_\mathcal{T}\}$. Neural Process variants achieve by maximising the marginal log-likelihood over the targets given a context set during training. Then, at test time, only the contexts are known and are used to generate samples at target points given the context and the conditional model.

This comes with **two** main issues:

**(a)**    Firstly, inferring from a conditional generative model sacrifices of consistent marginal distributions with respect to the context set $\mathcal{C}$, as it breaks conditional consistency (Xu et al., 2023). Hence, all previous NPs are at most (marginally) consistent *w.r.t* to the target set.

---

[1]See E.3 for a more comprehensive overview of the training procedure

**(b)** Secondly, at test time the context set can be vary over any distribution on the domain $X$ and length. Due to the many degrees of freedom, all these bases are hard to cover during training, this can lead to certain context set types not being well trained which can lead to inaccurate predictions.

With this in mind we propose using a conditional sampling method from the underlying unconditional model, similar to *guidance* Chung et al. (2024); Dhariwal & Nichol (2021) which is enabled due to the underlying diffusion model in *scoreNP*.

Neural Processes (Garnelo et al., 2018) define a conditional generative model by:

$$p(y_\mathcal{T}|x_\mathcal{T}) = \int_z q_\phi(z|\mathcal{C})p(y_\mathcal{T}|z, x_\mathcal{T})dz \quad (21)$$

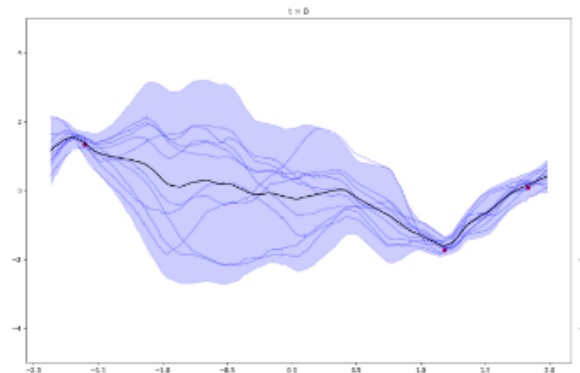

Figure 3: Posterior model distribution under conditional sampling given context points (red). Mean in (black), 2 std dev (blue) with individual samples overlayed

Where, $p(y_\mathcal{T}|\mathcal{C}, x_\mathcal{T})$ is a valid stochastic process and $q_\phi(z|\mathcal{C})$ is an approximate posterior. This approximation breaks conditional consistency (Xu et al., 2023), which implies they lose consistency *w.r.t* the context set and the underlying prior cannot be considered a stochastic process. However we can consider an equivalent conditional generative model, by using the fact that $q(z|\mathcal{C}) = p_\psi(z)p_\theta(\mathcal{C}|z)/p(\mathcal{C})$. rewriting Eq.21 as

$$p(y_\mathcal{T}|x_\mathcal{T}) = \int_z \frac{p_\psi(z)p_\theta(\mathcal{C}|z)}{p(\mathcal{C})}p(y_\mathcal{T}|z, x_\mathcal{T})dz \quad (22)$$

Eq.22 means conditional consistency will hold as the posterior is not approximated by the inconsistent encoder, $q_\phi(z|\mathcal{C})$

**Theorem 3** (Conditional Consistency). *Given a prior $p_\psi(f)$ over a stochastic process $f$, two arbitrary context sets $\mathcal{C}'$ and $\mathcal{C}$ with well defined conditional densities $p_\theta$. Furthermore, $p_\theta$ is consistent under marginalisation. Then the following equations hold. (see appendix A for proof)*

$$p_\psi(f) = \int p_\psi(f)p_\theta(\mathcal{C}'|f)d\mathcal{C}' = \int p_\psi(f)p_\theta(\mathcal{C}|f)d\mathcal{C} \quad (23)$$

This result implies that a generative model in the form of Eq.22 is fully consistent over the entire input domain and consistent *w.r.t* both context and target sets. We show how the diffusion model paradigm allows us use conditional generative model of the form Eq.22, which NPs have traditionally been unable to achieve.

**Conditional Diffusion**  We want to generate the conditional joint distribution $p_\theta^0(y_T, z \,|\, \mathcal{C}, x_\mathcal{T})$, which is the posterior under our score based diffusion at $\tau = 0$ conditioned on the context set. Hence, we want to approximate the conditional score for all $\tau \in [0, 1]$,

$$\nabla_{y_T^\tau, z^\tau} \log p^\tau(y_\mathcal{T}, z \,|\, \mathcal{C}) = \nabla_{y^\tau} \log p^\tau(y_\mathcal{T} \,|\, z) + \nabla_{z^\tau} \log p^\tau(z \,|\, \mathcal{C}) \quad (24)$$

$$= S_\theta(y_\mathcal{T}^\tau, x_\mathcal{T}, z^\tau, \tau) + \nabla_{z^\tau} \log p^\tau(z \,|\, \mathcal{C}) \quad (25)$$

Hence, we make an approximation for the conditional score of the latent distribution, $\nabla_{z^\tau} \log p^\tau(z \,|\, \mathcal{C})$.

**Estimating the conditional score**  As we want to sample from the latent conditioned on context, we consider the diffusion flow which will produce samples of $p^0(z|y_\mathcal{C}) = \frac{q^0(z)p^0(y_\mathcal{C}|z)}{p^0(y_\mathcal{C})}$. This implies we need to approximate the *conditional score* $\forall \tau \in [0, 1]$ :

$$\nabla_{z_\tau} \log \left[ \frac{p^\tau(z)p^\tau(y_\mathcal{C}|z)}{p^\tau(y_\mathcal{C})} \right] = \nabla_{z_t} \log p^\tau(z) + \nabla_{z_\tau} \log p^\tau(y_\mathcal{C}|z) - \underbrace{\nabla_{z_\tau} \log p^\tau(y_\mathcal{C})}_{\phantom{x}}{}^{\,0} \quad (26)$$

Leading to the following approximation for the conditional latent score.

$$\hat{S}_{\psi,\theta}(z^\tau, y_{\mathcal{C}}^\tau) \approx S_\psi(z_t, t) + \nabla_{z_t} \log \hat{p}_\theta^\tau(y_{\mathcal{C}}|z) \tag{27}$$

Where $\hat{p}_\theta^\tau(y_{\mathcal{C}}|z)$ is an approximation of the likelihood $p(y_{\mathcal{C}}^\tau|z)$ under a single denoising step in the (discretised) reverse process. Using the autograd function in PyTorch (Paszke et al., 2019) we can compute the gradients *w.r.t* $z^\tau$. In App.B we provide details on the approximation $\hat{p}_\theta^\tau(y_{\mathcal{C}}|z)$.

Similarly, to (Dhariwal & Nichol, 2021) we require scaling on the gradients of the likelihood to model accurate posteriors, choosing the scaling factor becomes a trade-off between diversity and underfitting. Extending the sampling to use more advanced sampling methods over simple predictor-only methods such as the recently proposed SMC based methods (Wu et al., 2023) for asymptotically exact posteriors is an exciting avenue for future research. This may alleviate the reliance on the troublesome guidance scale which is hard to tune and causes log-likelihood evaluation issues. see App.D.1

## 5 EXPERIMENTS

In this section we provide experimental validation of our model *scoreNP* on a range of function regression tasks. As *scoreNP* is a latent NP and a generalisation on NPs and MNPs we adopt the experimental set up from Xu et al. (2023). Predictive NPs such as convNPs, GNPs and (Geom) NDPs are not designed with full consistency in mind and hence forego certain constraints allowing for improved performance in predictive tasks.

Table 1: **1D Function Regression** All comparisons are lifted from table 1 in (Xu et al., 2023). Comparison models are as follows (a) Oracle models (when available). (b) GPs with optimised hyperparameters, including an RBF kernel, a Matern kernel and a periodic kernel. (c) CBPs which combine Gaussian copula processes with neural spline flows (Durkan et al., 2019) (d) NPs (Garnelo et al., 2018) (e) MNPs (f) ScoreNP is evaluated with an ELBO for two evaluations and no guidance scale is applied (details in D.1)

| Model | Samples from GP Kernels | | | Non-GP Data | | |
|---|---|---|---|---|---|---|
| | RBF | Matern | Periodic | Monotonic | Convex | SDEs |
| Oracle | $2.846 \pm 0.012$ | $2.709 \pm 0.013$ | $0.641 \pm 0.006$ | — | — | — |
| GPs | $\mathbf{2.844 \pm 0.013}$ | $\mathbf{2.708 \pm 0.014}$ | $0.419 \pm 0.013$ | $0.633 \pm 0.059$ | $2.976 \pm 0.224$ | $1.719 \pm 0.034$ |
| CBPs | $2.628 \pm 0.016$ | $2.604 \pm 0.015$ | $0.169 \pm 0.022$ | $1.776 \pm 0.088$ | $4.268 \pm 0.035$ | $1.842 \pm 0.024$ |
| NPs | $0.935 \pm 0.019$ | $1.115 \pm 0.021$ | $0.356 \pm 0.020$ | $1.823 \pm 0.006$ | $1.956 \pm 0.004$ | $1.621 \pm 0.009$ |
| MNPs | $2.491 \pm 0.024$ | $2.290 \pm 0.021$ | $\mathbf{0.594 \pm 0.032}$ | $2.755 \pm 0.010$ | $\mathbf{5.582 \pm 0.081}$ | $1.942 \pm 0.019$ |
| scoreNP | $2.532 \pm 0.04$ | $2.42 \pm 0.012$ | $0.0072 \pm 0.03$ | $\mathbf{3.801 \pm 0.030}$ | $4.033 \pm 0.050$ | $\mathbf{3.91 \pm 0.02}$ |

### 5.1 1D FUNCTION REGRESSION

We consider the controlled setting of 1D function regression problems. Three kernels (RBF, Matern, Periodic) are all considered where Oracle provides an upper bound on performance. In table 1 we show competitive performance over MNPs on the RBF and Matern but fall short of all comparisons in Periodic. We hypothesise that the higher frequency nature of this kernel results in a more complex which is currently not captured by our architecture. Across the non-GP datasets *scoreNP* shows improvements on both monotonic and SDE highlighting the ability of modelling complex distributions. The lower performance compared to CBPs and MNPs is likely due to the fact we don't use a guidance scale, this hypothesis is further validated as we see little improvement in the conditional log-likelihood to the unconditional (3.98 vs 4.03). 2D regression tasks and further qualitative results are shown in App. D.10 and App. B.1. We show failure cases of the model to generate coherent conditional samples given context on MNIST (LeCun & Cortes, 2010) desite strong unconditional likelihoods, which highlights the need for further analysis on architectural design.

## 6  RELATED WORK

**Infinite Dimensional Diffusion Models**    Diffusion Models have recently been extended to function spaces (Pidstrigach et al., 2023; Bond-Taylor & Willcocks, 2023; Biloš et al., 2023), theory of diffusion models to work in Hilbert spaces, which makes them theoretically well-defined. However, even with Neural architectures which heavily reduce inconsistency under discreation, such as Implicit Neural Representations (Sitzmann et al., 2020) or Neural Operators (Kovachki et al., 2021). This leaves a gap between practical and theoretical consistency (Phillips et al., 2022) decomposes the function space remaining marginally consistent to the target points but has limited representational power. (Mathieu et al., 2023; Dutordoir et al., 2023) both fully ignore consistency and work directly on finite dimensions, empirically showing that consistency is often approximated in the learning process. However, without the guarantees in place these models are still vulnerable to producing miscalibrated uncertainties and inconsistent predictions.

**Gaussian Processes and the Neural Process Family**    Gaussian Processes (GP) (Rasmussen & Williams, 2006) are perhaps the most important class of models for stochastic processes due to closed-form solutions and exact posteriors. However, they are extremely limited by the Gaussian assumption. The Neural Process Family is a neural approximate alternative to GPs, but also come with their own problems with consistencies and similar expressivity issues. Diffusion models provide a neat solution to these problems. (Bonito et al., 2023) use a the diffusion noising process to induce correlations for predictive NPs.

**Latent diffusion models**    have become a staple in image synthesis (Rombach et al., 2022) The benefit of the a latent prior learnt through a diffusion means you to sample from well supported regions of the latent space. This brings improved results in zero and low context settings where previous NPs have struggled. We followed a lot of (Vahdat et al., 2021) practical implementation for scoreNP. The training procedure of scoreNP is not dissimilar to that recently proposed in, *DisCo-Diff* (Xu et al., 2024),where they augment a diffusion model with discrete latents to induce smoother score networks.

## 7  CONCLUSION

**Limitations**    Sampling time of diffusion models is a drawback meaning the application high-speed sequential decision making isn't possible in the current scoreNP framework. Hence, improving synthesis speed is a necessary extension. The variational inference makes the learnability a difficult task with a couple of difficult balancing hyperparameters.

**Future work**    ScoreNP provides new avenues neural processes. Further in-depth empricial analysis on the training procedure to fully extract the power of diffusion models. The flexibility of non-strictly permutation invariant encoder may further improve the expressivity of the latent space orthogonal extensions to include inductive biases such as stationarity or equivariance is an exciting direction. The proposed conditional sampling opens up some exciting areas for future NP research.

**Conclusion**    We introduce *score-based Neural Process* (scoreNP), the first diffusion model which correctly adopts the NP framework satisfying the marginal consistency and exchangeability conditions of KET. We demonstrate how to train our model with a VLB based on a variational score-matching objective. We also show that leveraging reconstruction guidance for conditional sampling makes *scoreNP* the first NP variant to achieve conditional consistency. Our experiments on both unconditional and conditional function regression tasks quantitatively and qualitatively validates our framework.

## 8  REPRODUCIBILITY STATEMENT

We provide Algorithms in the main text 1 and the appendix 32 for the main algorithms which have novel aspects in our work. In D we provide details on how our datasets are collected and in E.3 we provide

an overview to our low level training procedure. In A we provide at least a sketch proof for all theorems proposed in the paper. If the paper is accepted, will provide open source code.

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

## A    PROOFS

In this section we will rewrite the proposed theorems and their corresponding proofs, including any required assumptions which were overly cumbersome for the main text.

**Theorem 1.** *Given initial value conditions that $z^1 \sim p^1(z)$ is tractable and $y_{1:n}^1 \sim p^1(y_{1:n}|x_{1:n})$ is tractable and defines a valid stochastic process over $X$, then $\forall \tau \in [0,1]$ the probability distribution $p^\tau(z)p^\tau(y_{1:n}|z, x_{1:n})$ determined by Eq.14 is a valid stochastic process over $X$*

*Proof.* (sketch proof) To prove this we will use the fact that the corresponding probability flow ODE to the SDE Eq.14, has identical marginals for all $\tau \in [0,1]$. Under a discretisation of this probability flow ODE, each step can be considered a Functional markov transition operator (FMTO) Xu et al. (2023). As $z^1 \sim p^1(z)$ is tractable and $y_{1:n} \sim p^1(y_{1:n}|x_{1:n}, z)$ with $p^1(y_{1:n}|x_{1:n}, z)$ being a valid stochastic process. Hence, at any time $\tau$ under discretisation the probability flow ODE produces consistent and exchangeable marginals.

By the strong Convergence theorem of Euler-Maruyama the marginals extend to being marginally consistent and exchangeable on the continuous time domain $[0, 1]$.

Finally, as the marginals under the probability flow ODE are equivalent to the SDE this completes the proof. □

**Theorem 2** (Variational Lower Bound)**.** *Under mild regularity conditions, the NLL over the data-distribution is upper bounded by:*

$$-\mathbb{E}_{p_{data}(y_{1:n})}\left[\log p_\theta(y_{1:n})\right] \leq J^{var\text{-}sm}(\theta, \psi, \phi; g(\cdot)^2) + \mathcal{H}_\phi(z) \tag{28}$$

*Proof.* Assumptions adapted from (Song et al., 2021a) to generalise to the joint setting:

(i) $p(x) \in C^2$ and $\mathbb{E}_{x \sim p}\left[\|x\|_2^2\right] < \infty$.

(ii) $p^1(x) \in C^2$ and $\mathbb{E}_{x \sim p^1}\left[\|x\|_2^2\right] < \infty$.

(iii) $\forall \tau \in [0,1] : f(\cdot, t) \in C^1, \exists C > 0, \forall x \in \mathbb{R}^D, t \in [0,1] : \|f(x,t)\|_2 \leq C(1 + \|x\|_2)$.

(iv) $\exists C > 0, \forall x, y \in \mathbb{R}^D : \|f(x,t) - f(y,t)\|_2 \leq C\|x - y\|_2$.

(v) $g \in C$ and $\forall \tau \in [0,1], |g(\tau)| > 0$.

(vi) For any open bounded set $O$, $\int_0^1 \int_O \left( \|p_t(x)\|_2^2 + \exists g(t)^2 \|\nabla_x p_t(x)\|_2^2 \right) dx dt < \infty$.

(vii) $\exists C > 0, \forall x \in \mathbb{R}^D, t \in [0,1] : \|\nabla_x \log p_t(x)\|_2 \leq C(1 + \|x\|_2)$.

(viii) $\exists C > 0, \forall x, y \in \mathbb{R}^D : \|\nabla_x \log p_t(x) - \nabla_y \log p_t(y)\|_2 \leq C\|x - y\|_2$. Where $p$ is the joint density

(ix) $\exists C > 0, \forall x \in \mathbb{R}^D, \tau \in [0,1] : \|s_\theta(x,t)\|_2 \leq C(1 + \|x\|_2)$.
$\exists C > 0, \forall x \in \mathbb{R}^D, \tau \in [0,1] : \|s_\psi(x,t)\|_2 \leq C(1 + \|x\|_2)$.

(x) $\exists C > 0, \forall w, z \in \mathbb{R}^D : \|s_\psi(w,\tau) - s_\psi(z,\tau)\|_2 \leq C\|w - z\|_2$.

(xi) $\exists C > 0, \forall z, x, y \in \mathbb{R}^D : \|s_\theta(x,z,\tau) - s_\theta(y,z,\tau)\|_2 \leq C\|x - y\|_2$.

(xiii) Novikov's condition: $\mathbb{E}\left[ \exp\left( -\frac{1}{2} \int_0^T \|\nabla_x \log p_t(x) - s_\theta(x,t)\|_2^2 dt \right) \right] < \infty$.

(xiv) $\forall \tau \in [0,1] \exists k > 0 : p^\tau(y_{1:n}, z) = O(\exp(-\|(y_{1:n}, z)\|_2^k))$ as $\|(y_{1:n}, z)\|_2 \to \infty$.

(xv) $\exists C > 0$, such that $z \sim q_\phi(\cdot), |z| < C$

We follow similar steps to what they proved in (Song et al., 2021a) Theorem 1 and Corollary 1. The first two steps are identical, let us define $\mu$, and $\nu$ to be path measures for two stochastic processes $\{U_t\}_{t \in [0,T]}$ and $\{\hat{U}_t\}_{t \in [0,T]}$

1.

$$D_{KL}\left( p_{data}(y) q_\phi(z|y) \,\middle|\middle|\, p_\psi^0(z) p_\theta^0(y|z) \right) \leq D_{KL}(\mu||\nu)$$

Due to data processing inequality.

2.

$$D_{KL}(\mu||\nu) \leq D_{KL}(p^1(y) q_\phi^1(z|y)||\pi_z \pi_{y|z}) + \mathbb{E}_{u \sim p^1(y) q_\phi^1(z|y)}\left[ D_{KL}(\mu(\cdot|U(1) = u)||\nu(\cdot|\hat{U}(1) = u) \right]$$

(29)

Due to the chain rule of KL divergence;

3. Using Girsanov's Theorem we get the following;

$$D_{KL}(\mu(\cdot|u(1)=x)||\nu(\cdot|u(1)=x)) \tag{30}$$

$$= -\mathbb{E}_\mu\left[\log\frac{d\nu}{d\mu}\right] \tag{31}$$

$$= \mathbb{E}_\mu\left[\int_0^1 g(t)(\nabla_u\log p_t(y)+\nabla_u\log q_\phi^t(z|y)-s_\theta(y,z,t)-s_\psi(z,t))d\bar{B}_t\right. \tag{32}$$

$$\left.+\frac{1}{2}\int_0^1 g(t)^2\left\|\nabla_u\log p^t(y)+\nabla_u\log q_t^\phi(z|y)-s_\theta(y,z,t)-s_\psi(z,t)\right\|_2^2 dt\right] \tag{33}$$

$$= \mathbb{E}_\mu\left[\frac{1}{2}\int_0^1 g(t)^2\left\|\nabla_u\log p_t(y)+\nabla_u\log q_t^\phi(z|y)-s_\theta(y,z,t)-s_\psi(z,t)\right\|_2^2 dt\right] \tag{34}$$

$$= \frac{1}{2}\int_0^1 \mathbb{E}_{p_t(y)q_t^\phi(z|y)}\left[g(t)^2\left\|\nabla_u\log p^t(y)+\nabla_u\log q_\phi^t(z|y)-s_\theta(y,z,t)-s_\psi(z,t)\right\|_2^2\right]dt \tag{35}$$

$$= \mathcal{J}^{var-sm}(\phi,\theta,\psi;\ g(\cdot)^2) \tag{36}$$

As this minimising KL is equivalent to maximising the log-likelihood this completes the proof. $\qquad\square$

**Theorem 3** (Conditional Consistency). *Given a prior $p_\psi(f)$ over a stochastic process $f$, two arbitrary context sets $\mathcal{C}'$ and $\mathcal{C}$ with well defined conditional densities $p_\theta$. Then the following equations hold.*

$$p_\psi(f) = \int_{\mathcal{C}'} p_\psi(f)p_\theta(\mathcal{C}'|f)d\mathcal{C}' = \int_{\mathcal{C}} p_\psi(f)p_\theta(\mathcal{C}|f)d\mathcal{C} \tag{37}$$

*Proof.* We take one of the arbitrary context sets $\mathcal{C}$.

$$\int_{\mathcal{C}'} p_\psi(f)p_\theta(\mathcal{C}'|f)d\mathcal{C}' = p_\psi(f)\int_{\mathcal{C}'} p_\theta(\mathcal{C}'|f)d\mathcal{C}' = p_\psi(f) \tag{38}$$

As we have a well-defined posterior likelihood $p_\theta$ over the context given the underlying stochastic process which induces consistent marginals on $\mathcal{C}$ we can marginalise over it. This process is identical for any context set which completes the proof.

$\qquad\square$

# B CONDITIONAL SCORE APPROXIMATION

Here we derive an approximation to the term $\nabla_{z^\tau}\log p^\tau(y_\mathcal{C}|z)$.

As we don't have an official decoder/classifier which is often used in this case, we use a single step under the learnt data diffusion to compute the likelihood approximation of $\hat{p}^\tau(y_\mathcal{C}|z) \approx p_\theta^{\tau|\tau+d\tau}(y_\mathcal{C}|z^{\tau+d\tau}, y_\mathcal{C}^{\tau+d\tau})$. The reverse SDE for a single step is normally distributed with variance $g(t)\sqrt{d\tau}$ through time reversal of Brownian motion. We set the mean to be the mean of $p^{\tau|0}(y_\mathcal{C})$ which we call $\mu^\tau(y_\mathcal{C})$. Now, as $\hat{p}$ Gaussian we can compute the closed form score. which is;

$$\nabla_{z^\tau}\log\hat{p}_\theta^\tau(y_\mathcal{C}|z) = -(\hat{y}_\mathcal{C}-\mu^\tau(y_\mathcal{C}))\nabla_{z^\tau}S_\theta(y_\mathcal{C}^{\tau+d\tau}, z^\tau, \tau, x_\mathcal{C})$$

## B.1 POSTERIOR DISTRIBUTIONS EXAMPLES

Here we showcase some unconditional and conditional model distributions from our trained ScoreNP on 1D datasets.

Periodic Kernel unconditional samples:

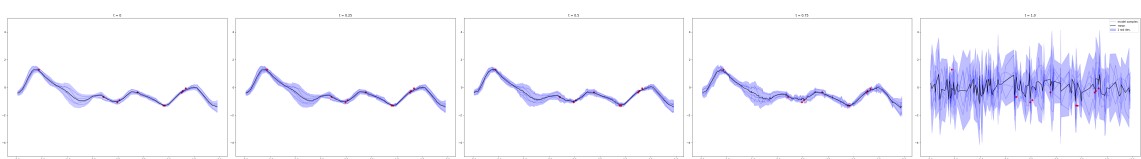

Figure 4: Results of conditional generation process for RBF Kernel with scaling parameter $s = 100\tau$.

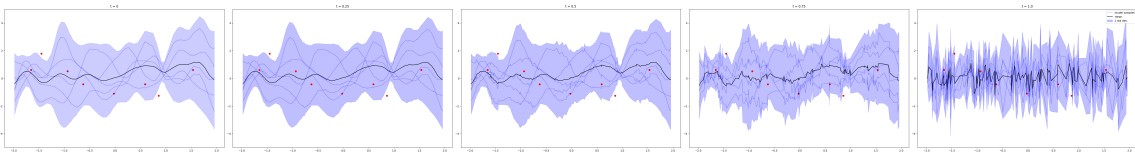

Figure 5: Results of conditional generation process for RBF Kernel with scaling parameter $s = 5\tau$, The small scale factor clearly shows inability to sample from a true posterior

## C  TRAINING AND SAMPLING ALGORITHMS

In this section we will outline the 3 main algorithms used throughout this work for training and sampling.

---

**Algorithm 2** Unconditional Sampling

---

**Require:** Score models $S_\theta(\cdot)$, $S_\psi(\cdot)$,
**Require:** Initial Distributions: $p^1(z) = \mathcal{N}(0, I)$, $p^1(y_i) = \mathcal{N}(0, I)$
**Require:** Terminal time $\epsilon$, time step $d\tau$
**Require:** SDE dynamics; $d(y, z)$
**Require:** Target input locations: $x_{1:n}$
  $T = 1$
  $z \sim p^1(z)$
  $y_{1:n} \sim p^1(y_{1:n})$
  **while** $T > \epsilon$ **do**
    Compute Latent Score: $S_\psi(z, T)$
    Compute Data Score: $S_\theta(y_{1:n}, z, T, x_{1:n})$
    $(z, y_{1:n}) := (z, y_{1:n}) + d(y, z)$
    $T = T - d\tau$
  **end while**
  Return Targets $(y_{1:n}, x_{1:n})$

---

Unconditional (Alg.2) and conditional sampling (Alg.3) are very similar, the operations on the targets remain the same, for conditional sampling we have additional requirements of context pairs and the forward transition density.

## D  DATASETS & EXPERIMENTS

Information about the datasets used and the likelihood evaluation which was adopted for all experiments.

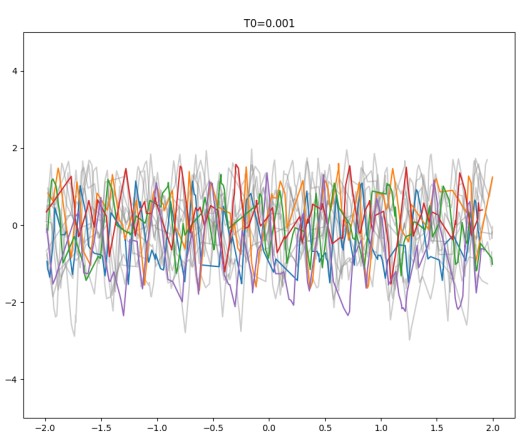

Figure 6: Unconditional samples periodic kernel

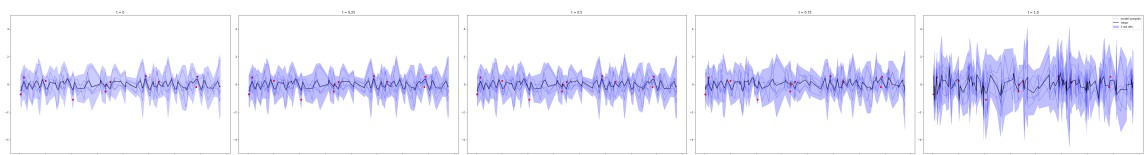

Figure 7: Posterior Distribution of Periodic Kernel under time reversal with $s = 200\tau$, this posterior shows clear underfitting

### D.1 LIKELIHOOD EVALUATION

We want to evaluate our model through log-likelihoods. Due to the variational aspect of our model we will only be able to approximate using NELBO. As mentioned in (Vahdat et al., 2021) IWAE objective often used in variational models can be biased due to variance when computing the likelihoods through the probability flow ODE. Hence, we resort to NELBO, which is:

$$\log p(y_{\mathcal{T}}|x_{\mathcal{T}}, y_{\mathcal{C}}) \geq \log p_{\theta,\psi}(y_{\mathcal{T}}, z|y_{\mathcal{C}}) - \log q_{\phi}(z|y_{\mathcal{T}}, x_{\mathcal{T}}) \tag{39}$$
$$= \log p_{\theta}(y_{\mathcal{T}}|z) + \log p_{\psi,\theta}(z|x_{\mathcal{C}}, y_{\mathcal{C}}) - \log q_{\phi}(z|y_{\mathcal{T}}, x_{\mathcal{T}}) \tag{40}$$

Where the the context set is a subset of the target set $\mathcal{C} \subset \mathcal{T}$ and $p_{\psi,\theta}(z|x_{\mathcal{C}}, y_{\mathcal{C}})$ denotes the latent density under conditional sampling given the context. If $\mathcal{C} = \emptyset$, $p_{\psi,\theta}(z|x_{\mathcal{C}}, y_{\mathcal{C}})$ simply becomes the unconditional model $p_{\psi}(z)$. The first two terms in 40 corresponding to the outputs of the under the joint diffusion are computed by solving the corresponding joint probability flow ODE (Song et al., 2021b; Grathwohl et al., 2018).

**No guidance scale** Although the guidance scale is useful for generating more accurate generations on the context points it is problematic for likelihood evaluation. Due to the gradients of the approximated likelihood $\hat{p}$ being untrustworthy, especially at times close to 0. This is due to the fact that the parametrised score model cares primarily about the value of $y$. This causes the latents to move outside of their true range and means that solving the probability ODE becomes infeasible with ODE solvers.

**Algorithm 3** Conditional Sampling using Guidance

---

**Require:** Score models $S_\theta(\cdot)$, $\hat{S}_{\psi,\theta}(\cdot)$,
**Require:** Initial Distributions: $p^1(z) = \mathcal{N}(0, I)$, $p^1(y_i) = \mathcal{N}(0, I)$
**Require:** Terminal time $\epsilon$, time step $d\tau$
**Require:** SDE dynamics; $d(y, z)$
**Require:** Transition density: $p^{\tau|0}(y_\mathcal{C})$
**Require:** Target input locations: $x_{1:n}$
**Require:** Context inputs and outputs: $(x_\mathcal{C}, y_\mathcal{C})$
   $T = 1$
   $z \sim p^1(z)$
   $y_{1:n} \sim p^1(y_{1:n})$
  **while** $T > \epsilon$ **do**
     Get Noised Context: $y_\mathcal{C}^T \sim p^{T|0}(y_\mathcal{C})$
     Compute conditional Latent Score: $S_{\psi,\theta}(z, T, y_\mathcal{C}^T)$
     Compute Data Score: $S_\theta(y_{1:n}, z, T, x_{1:n})$
     $(z, y_{1:n}) := (z, y_{1:n}) + d(y, z)$
     $T = T - d\tau$
  **end while**
  Return targets $(y_{1:n}, x_{1:n})$

---

## D.2 1D FUNCTIONS

We use the experimental set up from MNPs (Xu et al., 2023) and reproduce the dataset information here for the benefit of the reader.

## D.3 GAUSSIAN KERNEL FUNCTIONS

Variances were set at 0.0001 for the Radial Basis Function and Matern kernels, and 0.001 for the Exp-Sine-Squared kernel. Function Domain spanned from $-2.0$ to $2.0$. For this dataset, evaluation context size varies randomly from 2 to 50.

## D.4 MONOTONIC FUNCTIONS

The generation of monotonic functions starts by sampling $N \sim \text{Poisson}(5.0)$ to determine the number of interpolation nodes. We then sample $N + 1$ increments $X_{\text{increments}}$ sampled from a Dirichlet distribution. Increments are increased by 0.01 to avoid excessively small values, and are then normalized such that their sum is 4.0. The final $X$ values for interpolation nodes are obtained by adding $-2.0$ to the cumulative sum of these increments so that these $X$ values are within the range $[-2.0, 2.0]$.

For each $X$ value, a corresponding $Y$ value is sampled from a Gamma distribution $Y \sim \text{Gamma}(2, 1)$. The cumulative sum of $Y$ values ensures monotonicity. A PCHIP interpolator (Fritsch & Butland, 1984) is then created using these interpolation nodes ($X$ and $Y$ values) to generate function outputs. Given the functions, we randomly sample 128 $X$ values and compute their corresponding function values. Note that these $X$ values are now used to evaluate the functions, rather than serving as interpolation nodes. The function values are normalized to the range $[-1.0, 1.0]$. Finally, Gaussian observation noise with a standard deviation of 0.01 is added to these function values. For this dataset, evaluation context size varies randomly from 2 to 20.

### D.5 CONVEX FUNCTIONS

To create a dataset of convex functions, we compute integrals of the monotonic functions previously created. These convex functions are then randomly shifted and rescaled to increase diversity. The function values are normalized to the range $[-1.0, 1.0]$. Finally, Gaussian observation noise with a standard deviation of 0.01 is added to these function values. Evaluation Context sizes varied randomly from 2 to 20.

### D.6 STRATONOVICH STOCHASTIC DIFFERENTIAL EQUATIONS

We create a dataset of 1D functions, each of which represents a solution to a Stochastic Differential Equation (SDE). This SDE is defined by the drift function not to be confused with the SDE used for the diffusion model:

$$f(x, t) = -(a + x \cdot b^2) \cdot (1 - x^2)$$

and the diffusion function:

$$g(x, t) = b \cdot (1 - x^2),$$

with constants $a$ and $b$ both set to 0.1. The function sets up a time span that includes 128 uniformly distributed points within the range of $[-5.0, 5.0]$. We then uniformly sample an initial condition, $x_0$, between 0.2 and 0.6. We use the `sdeint.stratKP2iS` function from the `sdeint` library to generate a solution to the SDE. This solution forms a 1D function that depicts a trajectory of the SDE across the defined time span, originating from the initial condition $x_0$. Lastly, we randomly alter the eval context sizes between 2 and 50.

### D.7 1D DATASET SET SIZES

All of the 1D datasets have the same set sizes and splits: 50,000 for the training set, 5,000 for the validation set, and 5,000 for the test set.

### D.8 MNIST

We provide some results from experiments on MNIST (LeCun & Cortes, 2010). We achieve an unconditional ELBO of 7.1[2]. This is a large improvement on ConvCNP and AttnCNP (Gordon et al., 2020) which report a maximal log-likelihood of 1.27. in table 3. achieved by ConvCNPXL. We show that whilst we get, relatively good qualitative image samples from the unconditional model, the conditional generation does not succeed. Tuning the scaling parameter proved much harder in 2D to 1D.

### D.9 GEOLOGY

Similarly to MNIST, we applied our method to the geofluvial task introduced in (Xu et al., 2023). This task is a 2D function regression on grayscale images of river meandering on 128x128 images. The dataset is generated using (Sylvester et al., 2019). Our unconditional model reported log-likelihoods of 2.15. Whereas, MNP under maximal context size of 160 only achieved a marginal log-likelihood of $1.12 \pm 0.15$. We note that these results are not directly comparable, but MNP showed decreasing performance as the context decreases, perhaps suggesting that our unconditional model provides stronger likelihoods than MNPs context conditioned model.
Furthermore, the image synthesis we get from the unconditional model is not high quality in qualitative terms, see fig10 and 11 for comparison.

---

[2]we use $logp(y_{1:n}|z)$ as the estimate due to positively large latent likelihoods due to a very tight encoder

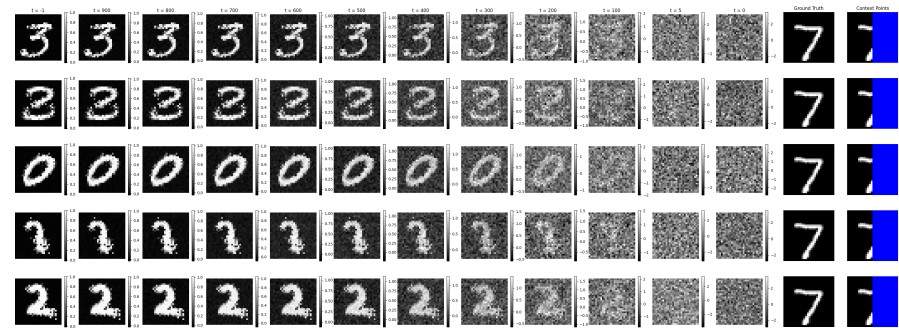

Figure 8: MNIST conditional genertion Samples given context (left)

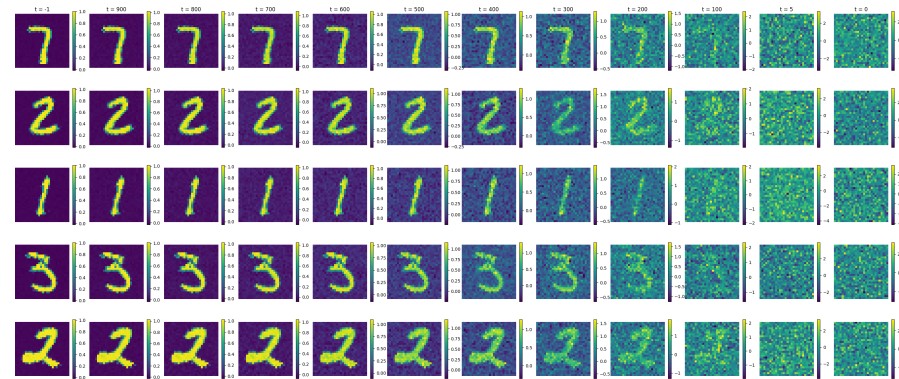

Figure 9: MNIST Unconditional generation Examples

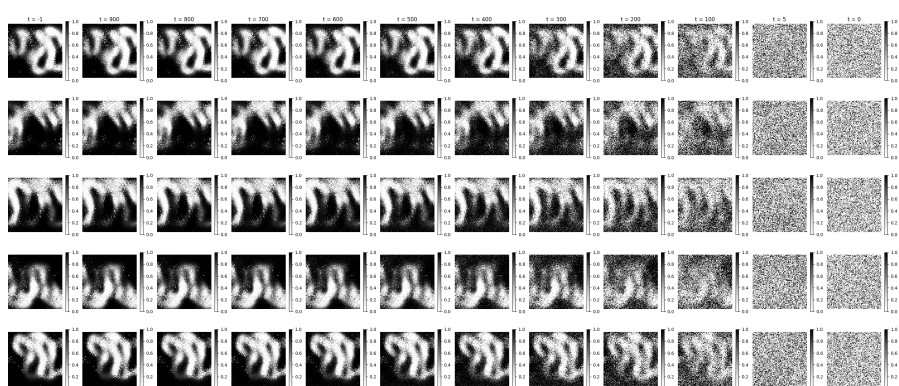

Figure 10: Geofluvial Unconditional Samples

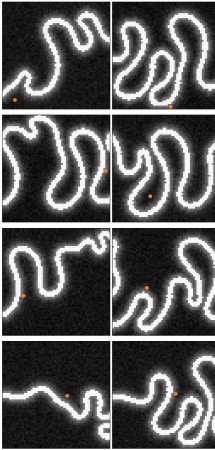

Figure 11: Ground Truth Geofluvial Examples

### D.10 QUALITATIVE RESULTS

## E IMPLEMENTATION DETAILS

### E.1 DIFFUSION SDEs, SAMPLING

**Geometric VPSDE** as mentioned, we adopt the use of a Geometric Variance preserving SDE for both the latent- and data-space. We provide the specific formulation of this process and the hyperparameters we used for our implementation. The forward process for a random variable $y$ under the Geo VPSDE.

$$dy^\tau = -\frac{1}{2}\beta(\tau)y^\tau d\tau + \sqrt{\beta(\tau)}dB^\tau \tag{41}$$

Where:

$$\beta(\tau) = \frac{\sigma_{min}^2(\frac{\sigma_{max}^2}{\sigma_{min}^2})^\tau}{1 - \sigma_{min}^2(\frac{\sigma_{max}^2}{\sigma_{min}^2})^\tau} \log \frac{\sigma_{max}^2}{\sigma_{min}^2} \tag{42}$$

The transition kernel is:

$$p^{\tau|0}(y) = \mathcal{N}\left(y^\tau; \sqrt{\frac{1 - \sigma_{min}^2(\frac{\sigma_{max}^2}{\sigma_{min}^2})^t}{1 - \sigma_{min}^2}}, \sigma_{min}^2(\frac{\sigma_{max}^2}{\sigma_{min}^2})^\tau \boldsymbol{I}\right) \tag{43}$$

We set $\sigma_{min}^2 = 3 \times 10^{-5}$ and $\sigma_{max}^2 = 0.995$. We found that setting $\sigma_{max}^2$ any higher resulted in unstable training and diverging samples.

The largeness of $g(\tau)^2$ as $\tau \to 1$ was ocassionally problematic and often caused instabilities in training, sampling and evaluation.

Using the VPSDE was a non starter, we experienced 0 learning signal even with importance sampling techniques suggested in (Vahdat et al., 2021)

**Sampling** We use 1000 uniform time steps of a predictor-only Euler-Maruyama sampler. This entails 2000 total NFEs across the latent-data diffusion. If we want to build a distribution over each sample in the batch using $n$ latent samples results in $(2n) \times 1000$ NFEs.

### E.2 ARCHITECTURES

Our model is made up of 3 main components during training,

1. Encoder: $q_\phi(\cdot|X, Y)$

2. Time-dependent Data Score: $S_\theta(y, t, x)$

3. Time-dependent Latent Score: $S_\psi(z, t)$

Once training is complete the Encoder is dropped and only the data-score and latent-score models are used during generation.

**Encoder** The encoder is a SetTransformer (Lee et al., 2019), DeepSet(Zaheer et al., 2018), or a Convolutional Encoder based on architectures in the Hugging Face diffusers library (von Platen et al., 2022) for grid based experiments (Images). The encoders take a collection of function input and output pairs $(X, Y)$ and output a mean and log-variance for a Normal distribution as in VAEs Kingma & Welling (2022). The mean is computed through the main encoder with a small DeepSet used to produce the log-variance. Additionally, the input locations $X$ are encoded with fourier features (Tancik et al., 2020).

**Time-dependent data score** The time-dependent score model conditions on the time, the latent representation at that time and pointwisely on the input location $x_i$ and the corresponding noised function output $y_i$. Initially, $z^\tau$ and $\tau$ are passed through a 1D Unet (von Platen et al., 2022), before employing a pointwise MLP conditioning on the Unet output and $(x_i, y_i)$. Each hidden layer of the MLP is appended with a layernorm (Ba et al., 2016).

**Time-dependent Latent Score** The latent variables are modelled as large 1D vectors. In the 1D regression tasks $z \in \mathbb{R}^{128}$. We use a 1D Unet (von Platen et al., 2022) from the diffusers library to model the scores.

### E.3 OPTIMIZERS, SCHEDULERS,

### E.4 COMPUTATIONAL REQUIREMENTS

Using an NVIDIA GeForce GTX 1080, training the 1D functions for 200 pretraining epochs (without the latent score) and 200 end-to-end epochs took approximately 12 hours. Generating samples required 2000 Neural function evaluations for 1000 predictor-only steps on the joint space, generating a batch of 128 samples took a couple of minutes.

