# OpenReview forum: "Score-Based Neural Processes"
_ICLR.cc/2025/Conference — ICLR 2025 Conference Withdrawn Submission_

### Official Review · Reviewer_Pk1A · 2024-10-21

**Soundness:** 1
**Presentation:** 1
**Contribution:** 1
**Rating:** 1
**Confidence:** 4

**Summary:**

The paper aims to develop more expressive neural process models by integrating score-based generative models, claiming enhanced handling of non-Gaussian stochastic process and the first achievement of conditional consistency within this framework. However, the submission is flawed with unsubstantiated claims, lack of rigorous theoretical support, and narrow experimental validation limited to 1D function regression tasks that do not conclusively outperform existing baselines.

**Strengths:**

Combining diffusion models with neural processes is a promising avenue in the study of neural processes.

**Weaknesses:**

* This paper contains many incorrect claims. Crucially, the development of the methodology and the associated proofs are fundamentally flawed. See detailed comments in the Questions section.

* Almost all figures are blurry and barely readable. Even significant zooming does not aid in understanding the content.

* The experiment design is far from satisfactory.  Only 1D function regression is used in this paper. Even for the 1D regression result, the proposed method doesn’t outperform existing baselines

* The presentation of the paper is highly problematic. The structure is disorganized and the language is unclear. For example, in the introduction part, the narrative chaotically oscillates between discussions on neural processes and diffusion models without a clear connection or transition. Additionally, the placement of the related work section after the experimental results is unconventional and disrupts the logical flow of the paper

**Questions:**

* The claim in line 18 that the proposed method is the first NP variant to maintain conditional consistency is incorrect. The first NP to achieve this was the Markov Neural Process (MNPs, Xu et al., 2023). The authors need to clarify how their approach to conditional consistency differs from or improves upon that of MNPs both theoretically and experimentally.

* Line 21, the paper fails to demonstrate that scoreNP outperforms existing neural process models based on the experimental results provided. 2D regression problems should be incorporated in the main text. Additionally, the emphasis should be on regression performance rather than generative capabilities.

* The organization of the presentation is disordered. The introduction starts with neural processes, shifts to diffusion models, discusses neural diffusion processes, and then returns to diffusion models.  Moreover, the relevance of Figure 1 to the function regression discussion is unclear.  The placement of the related work section after the experimental results is unconventional and disrupts the logical flow of the paper

* The claim in line 58 that operations directly on function space compromise consistency due to discretization is incorrect. Recent advances have shown that direct regression on function space can yield superior performance over existing NP baselines (Maronis et al., 2023; Shi et al., 2024).


* Lines 120-124 acknowledge that insights from MNPs inspired the development of an infinite chain of NPs via a denoising model. However, the assertion on line 229, which maps the point evaluation of an unknown stochastic process to $p^1(y_{1:n}|z, x_{1:n})$, namely $\mathcal{N}(0,I)$. This is fundamentally problematic because white noise doesn’t exist in L2 space.  You cannot define a stochastic process in L2 space such that joint distribution of a collection of points from this stochastic process is $\mathcal{N}(0,I)$. This issue has been extensively discussed in the literature on functional diffusion models (Lim et al., 2023; Kerrigan, 2023), where the Gaussian Process with trace-class operator is used instead of white noise.  Moreover, the proof of Theorem 1 is incorrect; the author incorrectly extends a finite-dimensional diffusion model defined on $\mathbb{R}^d $ to function space (specifically L2 space) by naively taking $d$ to infinity. This approach is flawed because density functions used in the finite-dimensional diffusion model do not exist in function space where the unknown stochastic process defined. A rigorous proof should start from the standpoint of measure theory.

* Line 591-591 “each step can be considered a functional markov transition operator of MNPs” is wrong.  In MNPs, the model uses normalizing flows, viewed as pointwise operators, applicable to stochastic processes observed at any position. Conversely, finite-dimensional diffusion model is restricted to $\mathbb{R}^d $ and cannot act as a valid operator in function space. To demonstrate that the proposed method is a valid operator, the author needs to provide related proof as well as zero-shot super-resolution experiments similar to those in functional diffusion models. (Lim et al., 2023; Kerrigan, 2023).

* The regression experiment is inadequate, with only 1D function regression tested against other baselines. Even within the 1D function results, there is no convincing evidence that the proposed method outperforms existing baselines, less alone the additional computation overhead required. More experiments are clearly needed, see the experiments in Dutordoir et al, 2023 for reference

## Reference
Jin Xu, Emilien Dupont, Kaspar Martens, Tom Rainforth, and Yee Whye Teh. Deep Stochastic Processes ¨ via Functional Markov Transition Operators, May 2023

Juan Maroñas, Oliver Hamelijnck, Jeremias Knoblauch, and Theodoros Damoulas. Transforming Gaussian Processes With Normalizing Flows. In Proceedings of The 24th International Conference on Artificial Intelligence and Statistics, pp. 1081–1089. PMLR, March 2021. URL https://proceedings.mlr.press/ v130/maronas21a.html. ISSN: 2640-3498

Yaozhong Shi, Angela F Gao, Zachary E Ross, and Kamyar Azizzadenesheli. Universal functional regression with neural operator flows. arXiv preprint arXiv:2404.02986, 2024.

Jae Hyun Lim, Nikola B. Kovachki, Ricardo Baptista, Christopher Beckham, Kamyar Azizzadenesheli, Jean Kossaifi, Vikram Voleti, Jiaming Song, Karsten Kreis, Jan Kautz, Christopher Pal, Arash Vahdat, and Anima Anandkumar. Score-based Diffusion Models in Function Space, November 2023. URL http://arxiv.org/abs/2302.07400. arXiv:2302.07400 [cs, math, stat].

Gavin Kerrigan, Justin Ley, and Padhraic Smyth. Diffusion Generative Models in Infinite Dimensions, February 2023. URL http://arxiv.org/abs/2212.00886. arXiv:2212.00886 [cs, stat].

Dutordoir, Vincent, Alan Saul, Zoubin Ghahramani, and Fergus Simpson. "Neural diffusion processes." In International Conference on Machine Learning, pp. 8990-9012. PMLR, 2023.

---

> ### Author Response · Authors · 2024-11-24
> **Thank you for review, response to first 5 Questions**
>
> We would like to thank the reviewer for their review and very thorough questions.
>
> We provide a response to all the questions in order in the response and take on board the constructive criticism regards to presentation of the paper. We first provide a response to the first 5 questions, with the rest in the next comment.
>
> - **Q1** Conditional Consistency is not achieved in MNP in practice [1], see the final paragraph in section 4. “In practice, the inference model $q_{x_{1:n}} (z^{(1:T )} | y_{1:n}; \phi)$ provides an approximate prior/posterior. Ideally, if it gives the exact posterior, conditional consistency would hold perfectly. However, when the inference model is approximate, the degree of conditional consistency depends on the discrepancy between the inference model and the true posterior."
> Moreover, any NP that uses a posterior $q(z| y_c, \phi)$, can by definition not be conditionally consistent. In ScoreNP, we form the posterior by a $p(z)p(y|z)$ which corresponds to an exact model posterior and therefore posteriors are generated from a single prior $p(z)$, satisfying conditional consistency. Hence, our model is the first NP variant to our knowledge to maintain conditional and marginal consistency.
>
>
> - **Q2** We understand the concerns with the empirical results, whilst the model doesn’t strictly outperform the existing NPs on GP regression it does have strong performance on 2 Non-GP datasets. We do only use an ELBO rather than IWAE or exact log-likelihoods, which may be a reason for the reduced performance, however, after finding an issue in our code we are unable to fully comment on the performance in the current form.
>
>
> -  **Q3** We thank the reviewer for the feedback on the organization of the work. We will work to improve on the writing in a revised version of the paper and apologise if it caused further effort during reviewing.
>
> - **Q4** This is not a claim to say that in practice the direct operation in the function space is flawed or doesn’t perform well. It is more a comment on the disparity between practice and theory when considering infinite-dimensional spaces, as one is required to perform discretisation in order to perform the operations in practice which does sacrifice the theoretical definitions. The empirical performance comparisons between function space operations and NPs is an interesting topic in its own right.
>
> - **Q5** To acknowledge the first part of this question where we refer the reviewer to sec. 4.3 in MNP [1]. “The initial SP can be arbitrarily chosen, as long as we can evaluate its marginals $p_{x_{1:n}}(y_{1:n}^{(0)})$. In our experiments, we use a trivial SP where all the outputs are i.i.d. standard normal distributed.”
> This implies that the initial samples for the finite marginals are drawn from $N(0,I)$ as is replicated by terminating the diffusion process at $N(0,1) = p^{1}$. Moreover, we are aware that in a Hilbert space $N(0,I)$ does not exist and don't claim otherwise. Similarly to GeomNDP [2] the use of finite dimensional marginals allows us to utilise white noise as a Target distribution. Moreover, one can define a process that terminates at a valid stochastic process in $L^{2}$. In [2], they provide different GP kernels as terminal distributions and conclude that $N(0,I)$ provides the best empirical results. Under the lebesgue measure finite marginal distributions defined $N(0_{n},I_{n})$ satisfy the Kolomogorov consistency conditions resulting in the underlying process being a valid stochastic process, however correctly not in the $L^{2}$.
> In the sketch proof, we used the fact that operating the probability flow ODE is equivalent to discrete transition in MNPs. We don’t naively extend $d \to \infty$, our finite dimensional diffusion also operates pointwise at any point of the stochastic process, providing a valid transition step from one finite dimensional marginal distribution to another without violating Kolomogorov Consistency conditions. This aligns the transition under our diffusion model to be equivalently a Markov Transition Operator introduced in [1]. We note that our diffusion process operates for any number of points $d$. The use of Measure Theory generally goes beyond the literature of Neural Processes which abstract the need of measure theory by the application of Kolomogorov Extension Theorem and operating on the finite marginals instead of directly in a function space. Due to the majority of the proof relying on Markov Transition Operators, we considered a sketch to be sufficient, however we will work on providing a more rigorous proof in a revised version of this work.
>
> References:
>
> [1] Jin Xu, Emilien Dupont, Kaspar Martens, Tom Rainforth, and Yee Whye Teh. Deep Stochastic Processes ¨ via Functional Markov Transition Operators, May 2023
>
> [2] Mathieu, E., Dutordoir, V., Hutchinson, M., De Bortoli, V., Teh, Y. W., & Turner, R. (2024). Geometric neural diffusion processes. Advances in Neural Information Processing Systems, 36.

---

> > ### Author Response · Authors · 2024-11-24
> > **Response to last 2 Questions**
> >
> > We provide further comment on the last 2 questions.
> >
> > - **Q6** This is a misinterpreted view on our model, specifically our diffusion model acts pointwise on any finite collection of points from the stochastic process. This implies under sampling, each discrete step is equivalent to a Markov Transition Operator [1]. Hence, the discrete sampling process becomes an equivalent to MNPs with the number of discrete steps determined by the sampling steps. Hence the operator is defined for the finite marginals with the output being a finite marginal distribution belonging to a valid stochastic process. .
> > The Zero-shot super-resolution tasks are an interesting experiment to run and we will look to include them in a revised version of this work, however, the performance on such a task would not validate or invalidate whether the model produces a valid stochastic process.
> >
> > - **Q7** We will look to include a more extensive experimental validation in a future version of this work.
> >
> > We hope that all the questions have been adequately answered and this alleviates the reviewers concerns with respect to the methodology. We would like to add that we have since found a bug in our code which will require further analysis in determining the effect of this on the empirical results.

---

> > > ### Comment · Reviewer_Pk1A · 2024-11-27
> > >
> > > Thank you for your response, I have reviewed the rebuttal and believe that significant improvements are needed for this paper to be accepted.  As such, I will maintain my score.

---

### Official Review · Reviewer_3jkT · 2024-10-29

**Soundness:** 1
**Presentation:** 1
**Contribution:** 2
**Rating:** 3
**Confidence:** 3

**Summary:**

This paper proposes Score-Based Neural Processes (scoreNP), integrating score-based generative models with Neural Processes (NPs) through joint diffusion in latent-data space. The authors claim two main contributions: (1) enhanced expressivity while maintaining marginal consistency through score-based modeling, and (2) achieving conditional consistency using guidance methods from conditional diffusion sampling. The method is evaluated on several 1D function regression tasks and image datasets.

**Strengths:**

* The combination of score-based models with NPs is an interesting direction. However, this combination has been explored in several works [1, 2, 3], the authors should discuss the advantages and differences of this work over previous ones.
* The joint latent-data space diffusion provides a theoretically motivated approach for handling both unconditional and conditional generation.
* The proposed solution for conditional consistency through guidance seems novel.


[1] Dou, H., Lu, J., Yao, W., qian Chen, X., & Deng, Y. Score-based Neural Processes.

[2] Dutordoir, V., Saul, A., Ghahramani, Z., & Simpson, F. (2023, July). Neural diffusion processes. In International Conference on Machine Learning (pp. 8990-9012). PMLR.

[3] Mathieu, E., Dutordoir, V., Hutchinson, M., De Bortoli, V., Teh, Y. W., & Turner, R. (2024). Geometric neural diffusion processes. Advances in Neural Information Processing Systems, 36.

**Weaknesses:**

The paper is still in very incomplete shape and requires substantial revision, I will list a few points:
* The paper is overall not well-written, with important details scattered across different sections, the language seems informal in places and the presentation is rushed and incomplete. Many technical terms are used without proper definitions, e.g., "Predictive NPs" and "Generative NPs", "Encoder collapse", "NFEs", "NELBO". Several equations contain unexplained variables and notations.
* The paper's theoretical development lacks rigor and clarity, particularly in explaining how score-based modeling integrates with Neural Processes. And Theorem 1's proof is incomplete.
* Critical implementation details are missing or insufficiently explained, e.g., appendix E.3 is empty; network architectures are vaguely described; Fourier feature encoding specifications are missing.
* The conditional sampling procedure's stability issues are not properly discussed.
* The experimental section is inadequate, which only runs some experiments on 1D synthetic functions, Appendix D.10 which should show results on 2D functions is empty. And there is a very limited comparison with state-of-the-art NPs. The failure cases on MNIST are concerning and not adequately analyzed.
* The font size in some figures is way too small. Figure 3 is too blurry, you should use vector graphics.
* The Related Work section is inadequate and misses several important connections and recent developments in NPs.
* The computational requirements section is vague.

I encourage the authors to enhance overall clarity and organization, add missing implementation details, and improve experimental validation.

**Questions:**

* What does encoder collapse mean?
* Can you give a more general introduction about "conditional consistency"?
* What causes the poor performance on periodic functions?

---

> ### Author Response · Authors · 2024-11-25
> **Thanks for review and feedback**
>
> We would like to thank the reviewer for their response and their extensive feedback on the presentation of this work. We will take on board your comments in a revised version of this work, particularly around the discussion of other similarly named works and the whole Neural Process literature.
>
> We also provide answers to the listed questions:
>
> - **Q1** Encoder Collapse, is when the output of the encoder becomes degenerate, generally $Enc_{\phi}(\cdot) \approx 0$. This means the latent space is degenerate so the Latent diffusion becomes trivial but the data-diffusion is unable to learn anything and generates noise. This can when the loss function has too large a weighting on the latent diffusion component.
>
> - **Q2** In our revised work, we will explain more the concept of Conditional Consistency and its impact on Neural Processes.
>
> - **Q3** We can postulate that it is due to the much less smooth nature of the functions in comparison to the others in the dataset, more extensive tuning of the architecture can hopefully improve this but we will not definitively comment on this as we have found a bug in the code which may be a reason for the weak performance. However, we agree it is definitely something that should be addressed in a revised version of this work.
>
> We thank the reviewers again for their feedback as it will be extremely useful in the construction of a revised version of this work and hope that the provided answers are adequate for the questions raised.

---

> > ### Comment · Reviewer_3jkT · 2024-11-27
> >
> > Thank you for your response to my questions. While I acknowledge the authors' efforts, I believe there remains substantial room for improvement in both the experimental validation and manuscript presentation. As such, I will maintain my current score.

---

### Official Review · Reviewer_aCJ3 · 2024-10-31

**Soundness:** 3
**Presentation:** 1
**Contribution:** 2
**Rating:** 3
**Confidence:** 4

**Summary:**

The authors introduce score-based neural processes (scoreNPs), a novel member in the neural process family that utilizes score-based generative modelling for enhanced expressivity. In comparison to other NPs, scoreNPs retain exchangeability and marginal consistency making them a valid stochastic process, and do not sacrifice consistency upon conditioning. The authors evaluate their method on several 1D regression tasks with mixed results.

**Strengths:**

- The paper proposes a promising method for the neural process family which definitely should be relevant to the community.
- The theoretical results are intriguing.

**Weaknesses:**

## Major
- The experimental section in the main text is extremely thin in comparison to the rest of the manuscript (some parts of the main text could in my opinion easily be moved to the appendix without hurting exposition in lieu of a more exhaustive experimental section).
- The method is in my opinion empirically not very convincing, e.g., it does not outperform the baselines on simple 1d regression tasks.
- The method seemingly fails to work on more complex tasks, such as unconditional generation of geofluvial data or conditional generation of MNIST.

## Minor
- The paper contains extremely many typos and grammatical errors, and sometimes misses entire words.
- The quality of the figures is low (pixelated, unreadable axes, ...).
- Some sections in the appendix seem to be empty, e.g. E.3.

**Questions:**

- The authors state: "The variational inference makes the learnability a difficult task with a couple of difficult
balancing hyperparameters."  Could they please elaborate to which parameters they are referring to and why they are difficult to tune?
- What is the advantage of this method is in comparison to, e.g., Markov NPs (leaving consistency aside)?
- The authors state (Appendix E.4.): "[G]enerating a batch of 128 samples took a couple of minutes." Is the method generally computationally as demanding?

---

> ### Author Response · Authors · 2024-11-24
> **Thank for review and response to questions**
>
> We would like to thank the reviewer for their time in providing a comprehensive review. We will take on the constructive criticism and work to improve the work in a revised version.
>
> We provide answers to the questions raised by the reviewer:
>
> - **Q1** We have since found a bug which potentially eradicates the issue of difficulty in the hyperparameters, but have yet to test extensively without it so cannot comment definitively. However, we were primarily referring to hyperparameters in the loss function for balancing the loss between score matching (data and latent) and the encoder entropy. Naively choosing a minimal loss can lead to the  latent score being perfectly learnt but the encoder collapsing.
>
> - **Q2** A couple of advantages are (i) the denoising score matching objective is generally an easier objective to optimise than Maximising Log likelihood (ii) having the latent space generated with a diffusion model itself means no matter what level of context is provided the sample will be generated from a well-supported region leading to strong results. This is akin to some of the improvements of Latent Diffusion Models over VAEs.
>
> - **Q3** No the method is not very computationally demanding, a couple of minutes was an overestimation of the sampling speed and was meant to imply that generation is rather quick yet will still be more expensive than MNPs or NPs which require much fewer sampling steps.
>
> We hope this answers the questions adequately and thank the reviewer again.

---

> > ### Comment · Reviewer_aCJ3 · 2024-11-27
> >
> > Thank you very much for addressing my comments and questions.
> >
> > I have read the others reviews which raised similar concerns and rebuttals, and will maintain my score.

---

### Official Review · Reviewer_ktGu · 2024-11-01

**Soundness:** 2
**Presentation:** 2
**Contribution:** 2
**Rating:** 5
**Confidence:** 2

**Summary:**

The paper presents Score-Based Neural Processes (scoreNP), a framework that enhances the expressivity and consistency of neural processes by integrating score-based generative models. scoreNP can model complex non-Gaussian distributions and generate correlated samples while maintaining both KET conditions.

**Strengths:**

The integration of score-based generative models within the NP framework is a novel approach. The theoretical foundation is robust- the proofs and methodology enhance trust in the results. The VLB objective is also a novel extension of likelihood re-weighting.

**Weaknesses:**

The paper only shows competitive performance for 1-D regression (that too only in certain cases) which raises questions about the scalability of the approach. Appendix D.10 (qualitative results sections) is empty. Although MNIST results for unconditional are competitive, the samples when conditioned on the context points are not good.

The authors acknowledge that tuning the architecture and hyperparameters gets very difficult even from 1-D to 2-D. This raises serious concerns about the scalability of this work.

**Questions:**

- Could the authors elaborate on how scoreNP would perform in higher-dimensional functional spaces? What modifications, if any, would be necessary to scale the approach?
- How sensitive is the model performance to the choice of hyperparameters?

---

> ### Author Response · Authors · 2024-11-24
> **Thank you for review**
>
> We would like to thank the reviewer for their response and respond their provided questions. We note that we have found a bug which will potentially fix the issues in training and scalability but we are yet analyse this to the full extent so will not make any definitive comments on the empirical performance.
>
> - **Q1** There are two avenues by which this task scales
> Increasing the dimension of the domain, i.e going from $1d$ to $2d$ gives input parameters of $(x1, x2)$ over $(x1)$. In many real physical processes this will be limited to $3d$ for $(x,y,z)$ coordinates and potentially $4d$ if time is also considered. This scaling would likely require the use of a higher-dimensional latent vector to encode the functions, with the encoder being free from any permutation invariances, at each of these scales the data diffusion is simply a flattened vector as the model operates pointwise. So any scalability limitations will primarily come from learning an effective latent embedding and diffusion model
> The other way of scaling is for the range of the functions to increase in dimensionality, for example RGB images $(y1, y2, y3)$ compared to grayscale $(y1)$. Scaling of this fashion would require greater capacity of the latent variable in the form of channels. This form of task scaling shouldn’t cause any issues in the scalability.
> The scalability aligns with that of latent diffusion models, in a revised version we will analyse and discuss this more in depth.
>
> - **Q2** The hyperparameters that were mentioned as issues are the ones in the loss function, balancing the losses of the latent diffusion, data diffusion and the entropy of the encoder does not directly transfer from task to task and  simply choosing the optimal parameters based off the lowest loss function can result in encoder collapse. We have since found a bug in the code which appears to stabilise this issue, but we have not had the chance to fully analyse this again. The model is not sensitive to other hyperparameters such as learning rates, other optimiser hyperparameters, architecture parameters etc.
>
> We hope this answers the provided questions adequately and thank the reviewer again for their response.

---

> > ### Comment · Reviewer_ktGu · 2024-12-02
> >
> > Thank you for responding to my questions. I will maintain my score.

---

### Note · Authors · 2024-12-19

**Comment:**

We are withdrawing the paper from consideration due to necessary changes and improvements still required. Thanks again to the reviewers, your insights will be very helpful in improving this work.

**Withdrawal Confirmation:**

I have read and agree with the venue's withdrawal policy on behalf of myself and my co-authors.